# Latent Classes of Adverse and Benevolent Childhood Experiences in a Multinational Sample of Parents and Their Relation to Parent, Child, and Family Functioning during the COVID-19 Pandemic

**DOI:** 10.3390/ijerph192013581

**Published:** 2022-10-20

**Authors:** Dylan Johnson, Dillon T. Browne, Robert D. Meade, Heather Prime, Mark Wade

**Affiliations:** 1Department of Applied Psychology and Human Development, University of Toronto, Toronto, ON M5S 1V6, Canada; 2Centre for Mental Health Research and Treatment, Department of Psychology, University of Waterloo, Waterloo, ON N2L 3G1, Canada; 3Human and Environmental Physiology Research Unit, School of Human Kinetics, University of Ottawa, Ottawa, ON K1N 6N5, Canada; 4Harvard T.H. Chan School of Public Health, Harvard University, Boston, MA 02115, USA; 5Department of Psychology, York University, Toronto, ON M3J 1P3, Canada

**Keywords:** adversity, benevolence, ACEs, BCEs, parental mental health, COVID-19

## Abstract

Adverse Childhood Experiences (ACEs) are known to contribute to later mental health. Conversely, Benevolent Childhood Experiences (BCEs) may buffer against mental health difficulties. The importance of ACEs and BCEs for mental health of both parents and children may be most obvious during periods of stress, with potential consequences for functioning of the family. Subgroups of ACEs and BCEs in parents during the COVID-19 pandemic were investigated and validated in relation to indices of parent, child, and family well-being. In May 2020, ACEs/BCEs were assessed in 547 parents of 5–18-year-old children from the U.K., U.S., Canada, and Australia. Subgroups of parents with varying levels of ACEs and BCEs were identified via latent class analysis. The subgroups were validated by examining associations between class membership and indices of parent and child mental health and family well-being. Four latent classes were identified: low-ACEs/high-BCEs, moderate-ACEs/high-BCEs, moderate-ACEs/low-BCEs, and high-ACEs/moderate-BCEs. Regardless of the extent of BCEs, there was an increased risk of parent and child mental health difficulties and family dysfunction among those reporting moderate-to-high levels of ACEs. Parents’ history of adversity may influence the mental health of their family. These findings highlight the importance of public health interventions for preventing early-life adversity.

## 1. Introduction

Adverse Childhood Experiences (ACEs) encompass a range of experiences and exposures such as physical, emotional, and sexual abuse, neglect, and household/community dysfunction occurring between the ages 0 and 17 years [1]. Considerable research has demonstrated the negative impact of ACEs on physical and mental health across the life course [2,3]. For example, ACEs are associated with problematic substance use, depression, anxiety, and post-traumatic stress [2,3,4]. In addition, parents with a history of ACEs are more likely to have children with behavioural difficulties, suggesting an intergenerational transmission of risk imbued by ACEs [5]. While the negative consequences associated with early adversity have been studied extensively, less is known about how ACEs co-occur alongside positive early-life experiences, also known as “Benevolent Childhood Experiences” (BCEs), which may have protective and promotive effects on development [6].

While ACE’s may impact the course of development and increase the risk for mental health difficulties in parents and their offspring, there is also evidence that ACEs may affect how adults are able to process stressful life events in the present, such as those related to the COVID-19 pandemic. For example, a study conducted by Doom et al. (2021) revealed that both ACEs and BCEs were associated with mental health during COVID-19 in a sample of university students, and that BCEs protected against mental health difficulties independent of ACEs [7]. Considering parents specifically, a meta-analysis conducted by Racine et al. (2022) demonstrated the significant impact that the pandemic has had on mental health of mothers with young children [8]. This is important given the critical role of parental well-being in supporting multiple dimensions of parent, child, and overall family functioning during the pandemic [9]. 

One widely adopted approach for the study of ACEs, and to a lesser extent BCEs, is the use of mixture modelling techniques such as latent class analysis (LCA). LCA is an exploratory, data-driven, and person-centred approach to identifying subgroups in a given sample, corresponding to distinct latent constructs (e.g., ACEs class), derived from multiple indicator variables (e.g., different types of ACEs) [10]. Cumulative models of traumatic events typically show a dose–response relationship with negative health outcomes [11], yet an assumption of these models is that all adversity items are equally weighted, thus losing potentially meaningful qualitative information about risks associated with specific types of adversity [12]. Since the influence of specific adversities may be different as compared to others, LCA provides a more nuanced approach for identifying classes of adversity co-occurrence and, ultimately, the potential to identify specific mechanisms for the development of psychopathology.

Multiple studies have applied LCA to identify distinct patterns of ACEs in a range of different populations. In samples of children, adolescents, and adults, a four-class solution is the most commonly observed [13,14,15,16,17,18], though there is evidence for solutions ranging from three to seven classes of differing qualities [19,20,21,22,23,24,25]. Fewer studies have applied LCA specifically to samples of parents. In one study of adolescent mothers by Stargel and Easterbrooks (2020), a four-class model was identified, with classes characterized as having low adversity, high abuse, high household dysfunction, and a multiple adversity group [26]. Relative to the other classes, the low adversity group had significantly better maternal and child health. Among the adversity-exposed classes, the high abuse group had lower child internalizing scores than the multiple adversity group, but comparable levels of child externalizing. In contrast, the high household dysfunction group had lower child externalizing scores than the multiple adversity group, but comparable levels of child internalizing. Nearly identical latent classes were identified in two other studies of adult mothers, which also identified increased odds of child maltreatment and intimate partner violence [27], as well as higher rates of postpartum depression [28], in the adversity-exposed classes relative to the low-adversity classes. 

Comparatively less research has applied mixture modelling to the measurement of BCEs and, perhaps due to the nascency of this measure, has primarily focused on validating the scale in its entirety (i.e., as an aggregate score). One study conducted both a cluster and confirmatory factor analysis in validating the BCEs measure in a community sample of Portuguese adults [29]. Beyond validating the BCEs construct in the confirmatory factor analysis, the cluster analysis revealed a three-cluster solution, characterized by groups of “Low-BCEs”, “Moderate-BCEs”, and “High-BCEs”. While a measure of adversity was not included in the cluster analysis, the study also showed that more ACEs were observed in the Low-BCEs group relative to the others, suggesting low BCEs and higher ACEs often co-occur. In another study of trauma-exposed patients in the UK, two confirmatory factor models were compared to determine if a single bipolar dimension better explained the ACEs and BCEs items than a model where these were treated as two distinct constructs [30]. Results provided support for the two distinct measures, suggesting these are not simply opposite ends of the same continuum. Lastly, in a study using structural equation modelling to study the effects of ACEs and BCEs on adult-reported family health, results showed that when accounting for BCEs, ACEs were associated with poorer family emotional and social health. In contrast, independent of ACEs, BCEs were positively associated with family emotional and social health, healthy lifestyles, health resources, and external social supports. Together, these studies suggest that not only are ACEs and BCEs capturing distinct dimensions of early experience, but they may have partially distinct health-related correlates. However, no study has applied LCA to model ACEs and BCEs items together or examined how the clustering of ACEs and BCEs are associated with parent, child, and family functioning during the COVID-19 pandemic.

### The Present Study

The goal of the current study was to identify latent classes from indices of ACEs and BCEs in a sample of parents with school-age children, and to validate these classes by measuring their association with parent mental health (anxiety, depression, post-traumatic stress, and substance use) and parenting behavior; child mental health (anger, anxiety, depression) and positive coping; and an overall measure of family functioning. Based on the exploratory and data-driven analysis of the current study, we did not make explicit hypotheses about the classes we expected to emerge; however, based on the extant literature, we hypothesized that there would be approximately four distinct classes, with one characterized by high co-occurrence of multiple adversities and one by overall low adversity [26,27,28]. In relation to the validation analyses, we hypothesized that classes characterized by high levels of ACEs and low levels of BCEs would be associated with more mental health difficulties of parents and their children, poorer parenting quality, and higher family dysfunction compared to classes characterized by low ACEs and high BCES. These hypotheses were pre-registered (https://osf.io/3sb57 (accessed on 27 March 2022)).

## 2. Materials and Methods

### 2.1. Study Design

This study was part of a larger longitudinal project that aimed to assess parent and child well-being during the COVID-19 pandemic, a period of heightened stress and disruption that may reveal individual differences in vulnerability to mental health difficulties and overall family well-being. The project was designed based on a conceptual model that explicitly underscores how pre-existing vulnerabilities (e.g., childhood adversity) may exacerbate mental health inequities and contribute to poor family adaptation during the COVID-19 pandemic [9]. The data were drawn from a longitudinal cohort of 549 parents and their first two children between 5–18 years of age (n = 1098) which began shortly after the COVID-19 pandemic began, with the first wave of data collection in May 2020. Participants were recruited through the research panel Prolific^®^, a third-party service that recruits participants for online research, with recruitment limited to parents with at least two children per family (aged 5–18 years), given the overarching study goals to explore within-family differences in mental health and well-being. Participants were recruited from the United Kingdom (76%), United States (19%), Canada (4%), and Australia (1%). Details on the overall study and primary findings are available elsewhere [31].

### 2.2. Participants and Procedure

After removal of two cases that were missing data on all ACEs and BCEs indicators, a total of 547 caregivers were included in the LCA and validation analyses. Constructs of interest included pandemic-related stress, family functioning, and mental health among parents and their children. Based on those with complete data, parents included in the analyses (n = 547) were a mean age of 41 years (*SD* = 6), majority female (70.08%), Caucasian (73.86%), non-migrants (89.52%), from two-parent households (90.50%), and made between $25–50,000 per annum (28.15%). The current study employed a cross-sectional design, analyzing data collected at the baseline assessment in May 2020 only. Parent-reported information on their history of ACEs and BCEs were used as indicator variables to define the latent classes in LCA, and data on parents, their children, and family functioning were used in validation analyses (described below).

### 2.3. Measures

Indicator variables. The indicator variables used in the LCA included multiple measures of ACEs and BCEs (items outlined in Appendix A). Participants retrospectively reported on their history of childhood adversity using the revised version of the Adverse Childhood Experiences Questionnaire [32]. For this measure, participants self-reported on 14-items pertaining to the presence or absence of childhood maltreatment and family dysfunction, including childhood abuse, neglect, peer victimization, exposure to community violence, socioeconomic status, etc. Participants also reported on their history of positive early life experiences using the Benevolent Childhood Experiences [6]. In the current study, the presence or absence of seven items pertaining to positive experiences including perceived safety, security, and support, were reported. 

Parental Anxiety. Anxiety was measured using the short-form of the Patient-Reported Outcomes Measurement Information System [33]. Participants responded to four items assessing fear, worries, and anxiety over the past seven days, with response options ranging from “Never” (1) to “Always” (5). Items were summed to generate a continuous measure of parental anxiety. Internal consistency in the current sample was good (Cronbach’s α = 0.93). 

Parental Psychological Distress. Psychological distress was assessed using the Kessler Psychological Distress Scale (K10), a widely used 10-item scale assessing the frequency of feelings related to depression and anxiety experienced in the past 30 days [34]. Response options ranged from “None of the time” (1) to “All of the time” (5). The continuous version of this scale was used and is the summation of all responses. Internal consistency in the current sample was good (Cronbach’s α = 0.93).

Parental Posttraumatic Stress. Post-traumatic stress was measured using the Trauma Screening Questionnaire [35]. Participants responded to a 10-item questionnaire assessing symptoms and behaviours pertaining to a trauma response, with example items including: “Upsetting thoughts or memories about the event that have come into your mind against your will”, “Feeling upset by reminders of the event”, and “Heightened awareness of potential dangers to yourself or others”. Item responses were coded on a binary scale (“Yes” = 1; “No” = 0). Items were summed to generate a continuous measure of parental posttraumatic stress. Internal consistency in the current sample was good (Cronbach’s α = 0.88).

Parental Substance Use. Substance use was assessed using the Tobacco, Alcohol, Prescription Medication, and other Substance use Tool [36]. Participants responded to 4 items assessing alcohol, tobacco, marijuana, and other prescription/non-prescription drug use in the past month. Response options ranged from “Everyday” (0) to “Never” (5). Items were reverse coded and summed to generate a continuous measure of parental substance use. Internal consistency in the current sample was acceptable (Cronbach’s α = 0.61).

Child Mental Health Problems. Parent-reported child mental health problems were captured using the PROMIS^®^. Specifically, the subscales of anger (5-items), anxiety (8-items), and depressive symptoms (6-items) were used in the current study [37]. Caregivers reported on five-point Likert scales with items ranging from “Never” (1) to “Almost Always” (5). Items were summed across subscales, with higher scores indicating greater mental health challenges. Internal consistency for this scale was good (Cronbach’s α = 0.94 for younger child and α = 0.94 for older child). 

Child Positive Coping. The 25-item Connor-Davidson Resilience Scale was utilized to assess child positive coping [38]. The measure includes items pertaining to adaptability, coping, hopefulness, and resourcefulness. Parents reported on behalf of their child on a five-point Likert scale with items ranging from “not true at all” (0) to “true nearly all of the time” (4). A sum score was generated whereby higher total scores reflected higher levels of positive coping. Internal consistency in the cohort was good (Cronbach’s α = 0.91 for younger child and α = 0.94 for older child).

Parenting Quality. The 10-item parenting practices scale from the 2014 Ontario Child Health Study was utilized to assess parenting quality [39]. Five of the items from the scale captured negative, coercive, and ineffective parenting, while the other five measured positive and supportive parenting. Parents reported on behalf of their child on a five-point Likert scale with items ranging from “Never” (1) to “Always” (5). Negative scores were reverse-coded so that the sum of the 10-item scale functioned where higher scores indicated higher parenting quality. Internal consistency in the cohort was good (Cronbach’s α = 0.81 for younger child and α = 0.83 for older child) [40].

Family Dysfunction. The 6-item General Functioning Subscale of the McMaster Family Assessment Device was utilized to assess family dysfunction [41,42]. The measure assesses familial aspects that include cohesion, planning, and support. Caregivers reported on a four-point Likert scale with items ranging from “strongly agree” (1) to “strongly disagree” (4) and were summed to generate total score, whereby higher scores indicated higher family dysfunction. Internal consistency in this cohort has been shown to be good (Cronbach’s α = 0.87).

Covariates. All validation analyses adjusted for parent age (years) at the time of data collection, sex (male, female), ethnic minority status (“yes” [1], “no” [0]), 2019 household income, immigration status (“yes” [1], “no” [0]), and single parent household status (“yes” [1], “no” [0]). 

### 2.4. Statistical Analysis

Latent Class Analysis. LCA was conducted using Mplus version 8 [43], while descriptive statistics and validation data analyses conducted using R studio software (version 4.0.2; packages outlined in Appendix A; discussed below). LCA is a mixture modelling technique whereby latent classes explain the relationship among a set of observed dependent variables, here referred to as indicators. When indicators are dichotomous, associations between indicators are quantitatively described by a set of logistic regression models. In the current study, a total of 21 dichotomous items were used as indicator variables (n = 14 ACEs items; n = 7 BCEs items; Appendix A). Solutions fitting between two and eight classes were compared, and guidelines for selecting the best-fitting model were used (Appendix A) [44]. The selection of the best-fitting model was determined by posterior fit indices that include bootstrapped likelihood ratio testing (BLRT) and Bayesian Information Criterion (BIC). In addition, model usefulness was determined by substantive interpretation and classification quality, determined by entropy, a standardised measure of the classification precision for placement of participants into identified classes derived from posterior probabilities, with values ranging from 0 to 1. Following identification of best fitting models, participants were assigned to the class in which they had the highest posterior probability of belonging to. 

Validation Analysis. For parental mental health, linear regression was conducted to examine differences in parental anxiety, depression, posttraumatic stress, and substance use between the latent classes. Classes identified in the LCA served as the independent variable. Based on the literature demonstrating that BCEs are associated with lower psychopathology and ACEs with higher psychopathology, the class characterized by high BCEs and low ACEs served as the reference group (‘Low-ACEs/High-BCEs’). Models were adjusted for all covariates. 

For child mental health, a model like the one described above was fit to data from the older child only. We had originally planned to perform a linear-mixed effects analysis including a nested random intercept (child within family); however, in preliminary analysis we observed that the models were not well-specified. We therefore opted for a simpler approach of examining associations with the older child only, and then performed sensitivity analyses where we fit the same model to data from the younger child, and to the average of the older and younger children. 

Missing Data. Missing values for covariates (i.e., age, sex, ethnic minority status, household income, and immigration status) were estimated via multivariate imputation by chained equations (MICE) using the MICE package in R. Predicted values were imputed via Bayesian linear regression (‘norm’ model, continuous variables), logistic regression (binary data), or polytomous logistic regression (unordered categorical data). Ten imputed data sets were generated and the results of each were analyzed individually and combined (Rubin’s Rules). Since data was collected using a completely anonymous survey, we assumed that self-reporting bias was minimal and that missing data could be assumed missing at random (i.e., the reason for missingness was not related to the outcome). The frequency of missing data is detailed in Appendix A and was generally low across variables.

## 3. Results

### 3.1. Latent Class Analysis

Fit indices supported a best-fitting model of four-classes (Appendix A). The proportion of participants reporting each ACE and BCE in the latent class group is depicted in Figure 1A,B, respectively. The patterns of adversity and benevolence differed across the classes as a function of severity. The first class was characterized by low levels of adversity and high levels of benevolence (Low-ACEs/High-BCEs; n = 270). The second class was characterized by moderate levels of adversity and high levels of benevolence (Moderate-ACEs/High-BCEs; n = 146). The third class was characterized by moderate levels of adversity and moderate levels of benevolence (Moderate-ACEs/Low-BCEs; n = 70). Finally, the fourth class was characterized by high levels of adversity and moderate levels of benevolence (High-ACEs/Moderate-BCEs; n = 61).

### 3.2. Descriptive Statistics

Table 1 presents the distribution of parent and older child outcomes and covariates by latent class group. Significant differences by latent class groups were observed for all variables except for ethnic minority status, income, and sex. Distributions of younger child outcomes are shown in Appendix A.

### 3.3. Validation Analysis

Parent Mental Health. Figure 2 depict raw distributions of parent mental health across LCA group, along with statistically significant, covariate-adjusted pair-wise comparisons. Significantly higher parental anxiety, psychological distress, and posttraumatic stress were observed in the Moderate-ACEs/High-BCE, Moderate-ACEs/Low-BCEs, and High-ACEs/Moderate-BCEs groups compared to the Low-ACEs/High-BCEs group. In addition, significantly higher levels of parental posttraumatic stress were observed in the High-ACEs/Moderate-BCEs group compared to the Moderate-ACEs/High-BCEs group. For parent substance use, significantly higher levels were observed in the Moderate-ACEs/High-BCEs and High-ACEs/Moderate-BCEs group compared to the Low-ACEs/High-BCEs group. Distributional assumptions (Appendix A), outcome correlation matrix (Appendix A), and unadjusted/adjusted multiple regression model results (Appendix A) are presented in the Appendix A.

Child Mental Health and Resilience. Figure 3A–D depict raw distributions of child mental health across LCA groups, along with statistically significant, covariate-adjusted pair-wise comparisons. For child anger, significantly higher levels were observed in the Moderate-ACEs/High-BCEs and Moderate-ACEs/Low-BCEs groups compared to the Low-ACEs/High-BCEs group. For child anxiety and depression, significantly higher levels were observed in the Moderate-ACEs/High-BCEs and High-ACEs/Moderate-BCEs compared to the Low-ACEs/High-BCEs group. In addition, significantly higher levels of anxiety were observed in the Moderate-ACEs/Low-BCEs compared to the Moderate-ACEs/High-BCEs group. No significant differences were observed between groups for the outcome of child positive coping. Distributional assumptions (Appendix A), outcome correlation matrices (Appendix A), and unadjusted/adjusted multiple regression models (Appendix A) are presented in the Appendix A. 

Family Functioning and Parental Quality Outcomes. Figure 4 depict raw distributions of family functioning and parenting quality across LCA groups. As compared to the Low-ACEs/High-BCEs group, all other groups showed significantly higher levels of family dysfunction. No significant differences were observed between the groups on parenting quality. Distributional assumptions (Appendix A), outcome correlation matrices (Appendix A), and unadjusted/adjusted multiple regression models (Appendix A) are presented in the Appendix A.

## 4. Discussion

The current study was an exploratory investigation into unique subgroups of parents with varying levels of childhood adversity (ACEs) and benevolence (BCEs), with a subsequent validation of these subgroups based on several metrics of family well-being during the COVID-19 pandemic, including parent and child mental health, family functioning, parenting behavior, and child positive coping. The results revealed four classes of ACEs/BCEs characterized by different overall severity levels of adversity and benevolence. Partially consistent with our initial hypothesis, classes characterized by higher levels of adversity had significantly higher levels of parental anxiety, posttraumatic stress, psychological distress, and substance use, as well as higher levels of child anger, anxiety, and depression, and higher levels of family dysfunction. Contrary to our hypotheses, no associations between the classes and either self-reported parenting or child positive coping were observed. 

Although the classes were generally differentiated by severity of ACEs/BCEs, there were also nuanced differences. Consistent with other research, one class in the current study was characterized by low levels of all adversity types, and another by high levels of all adversity types [26,27,28]. In contrast, the classes with moderate levels of adversity were characterized mainly by the presence of emotional and physical abuse, emotional neglect, and loneliness, but not the high levels of family dysfunction that were present in the group with high levels of all adversity (e.g., family member incarceration, household mental health, household substance use, low SES). These two classes being associated with most of the mental health difficulties observed is consistent with literature that has demonstrated an association between abuse, neglect, and subsequent development of mental health problems [2,3,4,45]. The fact that these two classes did not differ substantially on mental health outcomes from the class with high levels of all adversities (except posttraumatic stress), suggests that the interplay between emotional and physical abuse, emotional neglect, and loneliness may have a greater impact on mental health difficulties than adversities related to family dysfunction (which were only present in the class of high levels of all adversities) during the pandemic. While some literature shows an association between household dysfunction and mental health problems in mothers and their offspring [46], results on the topic are mixed [47], and more research is needed. 

Interestingly, the group with the highest levels of adversity did not have the lowest levels of benevolent experiences, consistent with the idea that ACEs and BCEs reflect two different dimensions of experience rather than simply occupying opposite ends of a single spectrum [30]. Moreover, the subgroup with the highest adversity did not differ in their risk of mental health difficulties relative to those exposed to moderate levels of adversity and higher levels of benevolence. This highlights two important findings. First, exposure to less complex (i.e., less varied) and moderate severity ACEs may pose a similar risk for mental health difficulties among parents as exposure to higher severity and more complex exposures; and second, there may be risks imbued by adversity that cannot be entirely mitigated by benevolent childhood experiences. Further evidence for this second point comes from the observation that there were still increased levels of all mental health difficulties in the Moderate-ACEs/High-BCEs group compared with the low-ACEs/High-BCEs group, which had comparable levels of BCEs. From a public health perspective, these findings highlight the need for preventative measures to guard against the experience of early adversity. However, it is also worth noting that this study was conducted during the COVID-19 pandemic, and we therefore cannot rule out the possibility that the combination of current pandemic stress and ACEs together confer such a burden that BCEs cannot adequately offset. Indeed, prior work suggests that childhood adversity and current pandemic stress may have additive and interactive effects in predicting parental mental health during the pandemic, and this double-burden may overtax parents’ capacity to mobilize coping resources to manage stress [48]. It may also be the case that *current* sources of social support and resilience, as opposed to sources of benevolence during childhood, are stronger buffers against mental health risk and poor family functioning during the pandemic. This requires additional follow up using validated scales for assessing pandemic stress and coping/adaptation.

This study also underscores that exposure to adversity, regardless of benevolent childhood experiences, was associated with increased levels of family dysfunction. This is partially consistent with findings by Daines et al. (2021) who showed that, when accounting for positive childhood experiences, childhood adversity was associated with reduced family social and emotional health processes and family health resources [49]. This may be especially important during periods of heightened stress and disruption―such as the COVID-19 pandemic―when the integrity and well-being of the entire family unit mediates the mental health and well-being of individual family members [9]. These results highlight one pathway by which parents’ own history of adversity may confer increased vulnerability for their children, namely by compromising the functioning and connectedness of the family, a hypothesis which has been supported by research during the pandemic [40]. 

Finally, no significant associations were observed between the LCA groups and either child positive coping or parenting quality. Research has shown ACEs to be associated with increased hostility to offspring, less sensitive and responsive parenting, and a higher proclivity towards the use of averse disciplinary techniques [47]. It may be that the effects of ACEs/BCEs during times of stress are more relevant for parental negativity or hostility, and that our measure was to crude to disambiguate this. Alternatively, the effects of ACEs/BCEs on parenting are less proximal to parents than their own mental health, and thus it may be that changes in mental health will eventually mediate effects on parenting, but that these changes may not have been as detectable during the earliest stages of the pandemic. Finally, for child positive coping, it may be that the findings reflect something about children’s own style of coping and adjustment to stress that is partly trait-like and partly environmentally mediated, but in either case may not be as strongly linked to parents’ own history of adversity, especially if their parenting (the environmentally mediated component) remains intact. 

### Strengths and Limitations

This investigation has several strengths. First, it is the first study to include both ACEs and BCEs in the same LCA model, and to further examine associations between derived classes and multiple indicators of parent, child, and family functioning during the COVID-19 pandemic. This approach provides insight into how experiences of adversity and benevolence are distributed in parents, as well as highlighting the need for preventative measures to reduce experiences of adversity that benevolent experiences may not fully buffer against. Second, analyses measuring the association between latent classes and validation outcomes were adjusted for important covariates that could confound the associations, adding precision to these estimates. Lastly, statistical rigor was used in adjusting for skewness of data in the LCA, applying multiple imputation to handle missing data, and correcting for heteroscedasticity in the outcomes using a heteroscedasticity consistent covariance matrix. Notwithstanding these strengths, there are also limitations. First, the utilization of a cross-sectional, observational design means that we are unable to ascertain the directionality of the relation between adversity, benevolence, and indicators of parent and child well-being. Thus, no causal conclusions can be drawn. Second, both ACEs and BCEs were reported retrospectively in adulthood, which may introduce recall biases. Indeed, prior research has demonstrated poor agreement between prospective and retrospective measures of childhood maltreatment, with higher rates being reported retrospectively [50]. This discrepancy was moderated by whether retrospective measures were collected via survey or interview, with the latter associated with higher agreement between retrospective and prospective measures. This suggests a need for replication of latent classes using prospective and/or interview-based measures of ACEs and BCEs. A third limitation is that participants were allocated into discrete LCA subgroups based on posterior probabilities, and proper class assignment cannot be guaranteed based on this approach (though classification accuracy was generally strong based on field-standard metrics). Due to restrictions on survey length, three BCEs items were not collected (“opportunities to have good time”, “liking oneself”, and “having a home routine”) and therefore not included in the LCA models. Due to the data-driven nature of the methodology, it is difficult to speculate how this may have impacted the findings. Replication with the full BCEs measure is therefore encouraged based on these preliminary results. Lastly, while the majority of the sample was recruited from the United Kingdom, a subset of the sample was recruited from the United States, Canada, and Australia. While the sample was not adequately powered to explore potential measurement invariance across the four nations, it is possible that sociocultural differences may introduce international discrepancies in the measurement, meaning, or interpretability of ACEs and BCEs or some of the validation outcomes. Future research should explore how cross-cultural and cross-national discrepancies impact the observed classes of ACEs and BCEs.

## 5. Conclusions

The current study is the first to include both measures of adverse and benevolent childhood experiences in the same LCA model to identify subgroups of parents who vary with respect to their early experience, and to link these classes to multiple domains of parent, child, and family function during the pandemic. The results are consistent with the proposal that, regardless of levels of childhood benevolence, even moderate exposure to a less complex range of childhood adversities is associated with parent and child mental health problems and more family dysfunction during the COVID-19 pandemic. While more research is needed, these findings suggest a need for interventions that target the prevention of adverse childhood experiences, or which aim to address its sequelae early in development to stave off the negative long-term risk associated with early adversity. 

## Figures and Tables

**Figure 1 ijerph-19-13581-f001:**
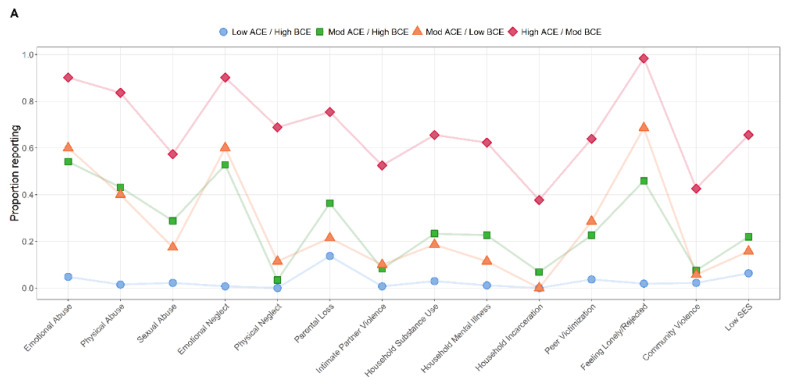
Item-level proportions of adverse childhood experiences (**A**) and benevolent childhood experiences (**B**) categorized by latent class analysis groupings. Note that although results are presented in two separate figures for ACEs (**A**) and BCEs (**B**), these were all included in the same model but are presented this way for ease of visualization.

**Figure 2 ijerph-19-13581-f002:**
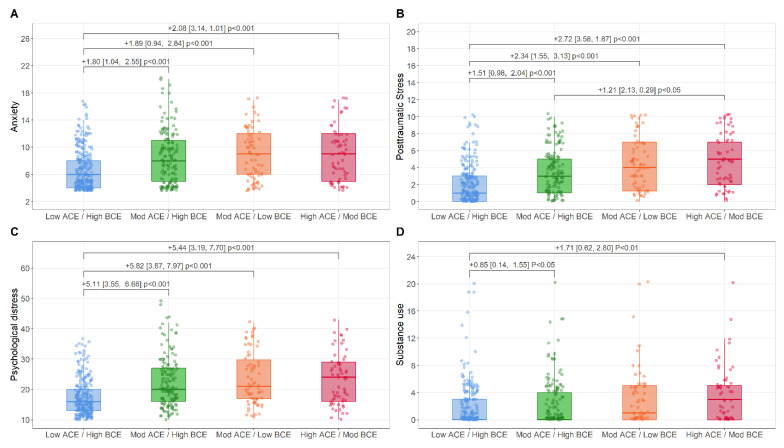
Boxplot representing raw scores of parent mental health validation outcomes of anxiety (**A**), posttraumatic stress (**B**), psychological distress (**C**), and substance use (**D**) across the LCA subgroups. Mean differences, confidence intervals, and *p*-values are adjusted for age, ethnic minority status, immigrant status, income, sex, and two parent household status.

**Figure 3 ijerph-19-13581-f003:**
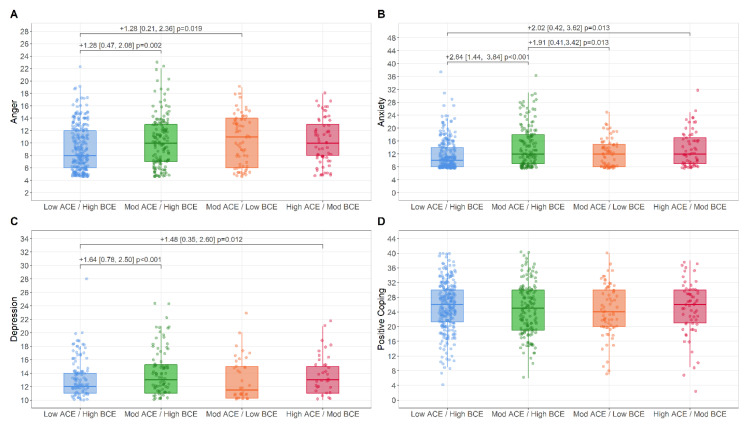
Boxplot representing raw scores of child mental health validation outcomes of anger (**A**), anxiety (**B**), depression (**C**), and positive coping (**D**) across the LCA subgroups. Mean differences, confidence intervals, and *p*-values are adjusted for age, ethnic minority status, immigrant status, income, sex, and two parent household status.

**Figure 4 ijerph-19-13581-f004:**
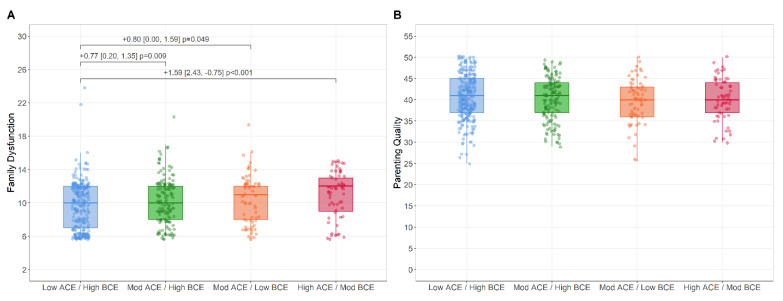
Boxplot representing raw scores of family dysfunction (**A**) and parenting quality (**B**) across the LCA subgroups. Mean differences, confidence intervals, and *p*-values are adjusted for age, ethnic minority status, immigrant status, income, sex, and two parent household status.

**Table 1 ijerph-19-13581-t001:** Distribution of mental health, positive coping, family functioning, parenting quality, and other sociodemographic factors among latent classes of adversity and benevolence.

	Low-ACEs/High-BCEs n = 270	Moderate-ACEs/High-BCE n = 146	Moderate-ACEs/Low-BCEs n = 70	High-ACEs/Moderate-BCEs n = 61	*p*-Value
Total ACES Count	0.42 (0.61)	3.77 (1.52)	3.69 (1.81)	9.54 (2.10)	*p* < 0.001
Total BCES Count	6.45 (0.80)	6.04 (0.95)	3.04 (1.06)	4.25 (1.76)	*p* < 0.001
Parent Distress	17.15 (6.06)	22.53 (8.40)	23.66 (8.71)	23.38 (8.39)	*p* < 0.001
Parent Anxiety	6.72 (2.99)	8.60 (4.03)	8.89 (3.77)	9.11 (3.94)	*p* < 0.001
Parent Posttraumatic Stress	1.92 (2.25)	3.48 (2.77)	4.36 (3.12)	4.84 (4.84)	*p* < 0.001
Parent Substance Use	1.66 (3.12)	2.43 (3.67)	2.93 (4.32)	3.61 (4.21)	*p* < 0.001
Family Dysfunction	9.51 (2.88)	10.27 (2.77)	10.39 (2.90)	11.08 (2.82)	*p* < 0.001
Child Anger	9.14 (3.66)	10.52 (4.02)	10.38 (4.10)	10.20 (3.73)	*p* < 0.001
Child Anxiety	11.73 (4.76)	12.38 (4.31)	14.32 (6.40)	13.62 (5.52)	*p* < 0.001
Child Depression	9.83 (3.72)	10.54 (3.79)	11.40 (4.37)	11.31 (3.96)	*p* < 0.001
Child Positive Coping	25.60 (6.67)	24.32 (7.05)	24.74 (7.08)	25.00 (7.40)	*p* = 0.450
Parenting Quality	40.70 (5.17)	39.74 (5.12)	40.21 (4.96)	40.25 (4.95)	*p* = 0.495
Age	42.1 (6.3)	40.8 (5.9)	40.7 (6.5)	39.8 (6.9)	*p =* 0.027
Ethnic Minority					*p =* 0.156
Yes	33 (14.0%)	22 (17.6%)	4 (6.9%)	4 (8.2%)	
No	202 (86.0%)	103 (82.4%)	54 (93.1%)	45 (91.8%)	
Immigrant					*p =* 0.002
Yes	28 (10.4%)	25 (17.2%)	3 (4.3%)	1 (1.6%)	
No	240 (89.6%)	120 (82.8%)	67 (95.7%)	60 (98.4%)	
Income					*p =* 0.095
<25 k	26 (9.6%)	22 (15.2%)	10 (14.3%)	11 (18.0%)	
25 k–49.99 k	69 (25.6%)	38 (26.2%)	30 (42.9%)	17 (27.9%)	
50 k–74.99 k	74 (27.4%)	31 (21.4%)	13 (18.6%)	15 (24.6%)	
75–99.99 k	51 (18.9%)	23 (15.9%)	9 (12.9%)	6 (9.8%)	
>=100 k	50 (18.5%)	31 (21.4%)	8 (11.4%)	12 (19.7%)	
Sex					*p =* 0.134
Female	172 (65.4%)	103 (74.6%)	50 (73.5%)	45 (76.3%)	
Male	91 (34.6%)	35 (25.4%)	18 (26.5%)	14 (23.7%)	
Two Parent Household					*p =* 0.047
Yes	253 (93.7%)	128 (87.7%)	63 (90.0%)	51(83.6%)	
No	17 (6.3%)	18 (12.3%)	7 (10.0%)	10 (16.4%)	

Data reported as mean (*SD*) or count (%). *p*-value represents results from a *t*-test (continuous variables) or chi-square test (categorical variables). All chi-square tests had expected cell counts >=5. Child variables displayed are for the older child (see Appendix A for younger children).

## Data Availability

Due to ethical constrictions on the sensitive data used in this investigation, the data cannot be shared. The data contain potentially identifying and sensitive data on a sample of parents and their offspring. In addition, REB # 42112 was reviewed by the Health Research Ethics Board at the University of Waterloo as the survey instruments contained sensitive questions. Participants were informed of the possibility that the data could be used for future research (in which the researchers would reach out to the Primary Investigator), and consent was collected for this. Participants who agreed to the use of their data for future research may feel that this is different than having their data being shared publicly.

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
