# Peer review of "Latent Classes of Adverse and Benevolent Childhood Experiences in a Multinational Sample of Parents and Their Relation to Parent, Child, and Family Functioning during the COVID-19 Pandemic"

_ijerph, 2022, doi:10.3390/ijerph192013581_

Round 1
Reviewer 1 Report
Dear authors,
The manuscript is well structured and written. It clearly explains a rather complex investigation.
The authors write that the results they obtain are not what they expected. They also explain the limitations of the design and the retrospective data. The way they do it is very commendable. I believe it can be very useful for students and future researchers.
Please see the attachment.

Author Response
Thank you for your insightful review of our manuscript. Your feedback has helped us improve and better clarify our research. Please find our reply to each of your suggestions below.
Reviewer Comment: Table S8. Distribution of younger child mental health, positive coping, and parenting quality, among latent classes of adversity and benevolence. Is on page 37, but not in Table of Contents. The authors should correct this.
Response: Thank you for identifying this, Table S8 is now visible in the Table of Contents.
Reviewer Comment: Authors should explain in their manuscript how they have selected the 14 items ACE and 7 BCE items. It seems that the original instrument BCE Scale has 10 items (Narayan, A. J., Rivera, L. M., Bernstein, R. E., Harris, W. W., & Lieberman, A. F. (2018). Positive childhood experiences predict less psychopathology and stress in pregnant women with childhood adversity: A pilot study of the benevolent childhood experiences (BCEs) scale. Child abuse & neglect, 78, 19-30.). What criteria have the authors followed for their selection? Is it a validated selection? Do they follow previous studies? Is it adapted to all populations: people from United Kingdom (76%), United States (19%), 144 Canada (4%), and Australia (1%)?
Response: During the initial study design, there were some scales that were shortened to reduce overall participant burden. As such, in this study, a shortened version of the BCEs measure was used. The omitted items included, “opportunities to have a good time”, “liking/feeling comfortable with yourself”, and “predictable home routines”. We acknowledge this limitation in the discussion by stating, "Due to restrictions on survey length, three BCEs items were not collected (“opportunities to have good time”, “liking oneself”, and “having a home routine”) and therefore not included in the LCA models. Due to the data-driven nature of the methodology, it is difficult to speculate how this may have impacted the findings. Replication with the full BCEs measure is therefore encouraged based on these preliminary results."
While this abbreviated 7-item scale has not been used by other research teams in the past, we recently published on this truncated BCEs scale (Prime, Wade, & Browne, 2022, Adversity & Resilience Science), with some evidence for its validity on the basis of its capacity to distinguish between risk groups during the pandemic. We also note that the measure had acceptable internal consistency reliability (α = .68), though for the current study it was the individual items used as opposed to the entire scale.
Reviewer Comment: On lines 157-161 they said: “For this measure, participants self-reported on 14-items pertaining to the presence or absence of childhood maltreatment and family dysfunction, including childhood abuse, neglect, peer victimization, exposure to community violence, socioeconomic status, etc. Participants also reported on their history of 160 positive early life experiences using the Benevolent Childhood Experiences [6]”. It would be important to better explain the difference between 7 BCE items (Table S1) and the 160 positive early life experiences.
Response: I believe this is an error in the sentence running into the line number (line # 160). The sentence should read, "Participants also reported on their history of positive early life experiences using the Benevolent Childhood Experiences [6]" and not "Participants also reported on their history of 160 positive early life experiences using the Benevolent Childhood Experiences [6]"
Reviewer Comment: Authors should explain how consider the level to be high, moderate, or low.
Response: We believe the Reviewer is referring to the labels given to the profile groups. These labels reflect qualitative distinctions based on the relative likelihood of endorsement of the items within each profile. Admittedly, this is a bit of a subjective enterprise but one that is common in studies using LCA in order to aid description and interpretability of the profiles. That said, we are open to other considerations should the Reviewer feel another labeling strategy is preferable.
Reviewer 2 Report
I would like to thank the opportunity of reviewing the manuscript entitled “Latent classes of adverse and benevolent childhood experiences in a multinational sample of parents and their relation to parent, child, and family functioning during the COVID-19 pandemic”. It exposes clearly different conclusions about childhood experiences during COVID-19 lockdown that are significant for the mental health of the minors in a long-term evaluation of the events.
I liked very much the method of the article and found it very interesting, with strong enough conclusions for the aim of the article.
However, there are a few issues that I would like the authors to consider.
P.3., l.119: When explaining the project design, I think this part would fit better in a Design section inside Materials & Method.
P.3., line 141: What the authors mean with “Participants were recruited through the research panel Prolific”?
Congratulations on their work.
Author Response
Dear Reviewer,
Thank you for your insightful review of our manuscript. Your feedback has helped us improve and better clarify our research. Please find our reply to each of your suggestions below.
Reviewer Comment: When explaining the project design, I think this part would fit better in a Design section inside Materials & Method.
Response: We have revised this section and moved the project design components from the “The Present Study” section into the “Materials & Methods” section. In addition, we have added a section into the “Materials & Methods” section which details the “Study Design” while also keeping a section devoted to “Participants and Procedure”.
Reviewer Comment: What the authors mean with “Participants were recruited through the research panel Prolific”?
Response: To better clarify what the research panel is, we have changed the text to read, “Participants were recruited through the research panel Prolific®, a third-party service that recruits participants for online research”.
Reviewer 3 Report
Let me say that I bumped into an excellent manuscript in all its parts. It is well-structured, the hypotheses are proper and sound, the latent class analyses are very well-conducted and the findings and thus conclusions are very insightful. The English is clear and fluent.
I have just two minor issues that need attention: 1) within the subheading 2.1 Participants and procedure on line 140 where details on the overall study were made available to the readers, I found the link: "https://uwaterloo.ca/whole-family-lab/current-research/cramped-families-study (accessed on 27 March 2022)" down with "page not found", so I do suggest to check this; 2) regarding the participants I could not understand why they came from different countries. I do reckon it has concerned a design of the initial study, but I do also think that it is important to spend some more words and explain if there were specific reasons behind. I also arguing that because whenever structured questionnaires with measurement scales are being assigned to respondents with different cultures, possible problems of measurement invariance may occur on the answers scored by means of Likert-scales.
Author Response
Dear Reviewer,
Thank you for your insightful review of our manuscript. Your feedback has helped us improve and better clarify our research. Please find our reply to each of your suggestions below.
Reviewer Comment: Within the subheading 2.1 Participants and procedure on line 140 where details on the overall study were made available to the readers, I found the link: "https://uwaterloo.ca/whole-family-lab/current-research/cramped-families-study (accessed on 27 March 2022)" down with "page not found", so I do suggest to check this.
Response: I think the issue here is that the “(accessed on 27 March 2022)” portion of the reference is a part of the reference formatting but should not be part of the survey link. This is unclear however, so we have opted to remove it from the reference. The link provided should not work properly: https://uwaterloo.ca/whole-family-lab/current-research/cramped-families-study
Reviewer Comment: Regarding the participants I could not understand why they came from different countries. I do reckon it has concerned a design of the initial study, but I do also think that it is important to spend some more words and explain if there were specific reasons behind. I also arguing that because whenever structured questionnaires with measurement scales are being assigned to respondents with different cultures, possible problems of measurement invariance may occur on the answers scored by means of Likert-scales.
Response: Thank you for raising this important point. The participants were recruited via a third-party service that recruits participants for online research. Due to the time sensitive nature of the pandemic, we opted to recruit from all available regions for which Prolific® surveyed participants to ensure we reached our desired participant number in a timely manner. There are of course benefits and drawbacks to this approach, with a primary limitation being that we cannot be sure that the questionnaire has the same meaning and interpretability across all countries. On the one hand, this concern was likely mitigated by the fact that all countries are similar in many respects culturally, socially, and economically, as well as the measure of adversity having been used in Australia (Tregeagle et al., 2019), Canada (McDonald et al., 2019), the U.K. (Bellis et al., 2014), and the U.S. (Finkelhor, 2020). On the other hand, there are fundamental differences between the countries in social policies and cultural practices that may impact certain dimensions of the current study. Unfortunately, our study was not powered to explore measurement variance across the four different nations. With such a small sample coming from Canada and Australia, it is not appropriate to test for differences in reliability of measurement across sites. This is a limitation that requires discussion which we now mention on page 13, where we say: “Lastly, while the majority of the sample were recruited from the United Kingdom, a subset of the sample was recruited from the United States, Canada, and Australia. While the sample was not adequately powered to explore potential measurement invariance across the four nations, it is possible that sociocultural differences may introduce international discrepancies in the measurement, meaning, or interpretability of ACEs and BCEs or some of the validation outcomes. Future research should explore how cross-cultural and cross-national discrepancies impact the observed classes of ACEs and BCEs.”
References
Bellis, M.A.; Lowey, H.; Leckenby, N.; Hughes, K.; Harrison, D. Adverse Childhood Experiences: Retrospective Study to Determine Their Impact on Adult Health Behaviours and Health Outcomes in a UK Population. J. Public Health. 2014, 36, 81–91, doi:10.1093/PUBMED/FDT038.
Finkelhor, D. Trends in Adverse Childhood Experiences (ACEs) in the United States. Child Abuse Negl. 2020, 108, 104641, doi:10.1016/J.CHIABU.2020.104641.
McDonald, S.W.; Madigan, S.; Racine, N.; Benzies, K.; Tomfohr, L.; Tough, S. Maternal Adverse Childhood Experiences, Mental Health, and Child Behaviour at Age 3: The All Our Families Community Cohort Study. Prev. Med. 2019, 118, 286–294, doi:10.1016/J.YPMED.2018.11.013.
Tregeagle, S.; Moggach, L.; Trivedi, H.; Ward, H. Previous Life Experiences and the Vulnerability of Children Adopted from Out-of-Home Care: The Impact of Adverse Childhood Experiences and Child Welfare Decision Making. Child. Youth Serv. Rev. 2019, 96, 55–63, doi:10.1016/J.CHILDYOUTH.2018.11.028.